Impact of Mycoplasma ovipneumoniae on juvenile bighorn sheep (Ovis canadensis) survival in the northern Basin and Range ecosystem

Spaan Robert S. rob.spaan@oregonstate.edu 1
Epps Clinton W. 1
Crowhurst Rachel 1
Whittaker Donald 2
Cox Mike 3
Duarte Adam 1 4
1 Department of Fisheries and Wildlife, Oregon State University , Corvallis , OR , United States of America
2 Oregon Department of Fish and Wildlife , Salem , OR , United States of America
3 Nevada Department of Wildlife , Reno , NV , United States of America
4 Pacific Northwest Research Station, USDA Forest Service , Olympia , WA , United States of America
Bolshoy Alexander
Electronic publication date: 2021 Jan 19
Publication date: 2021
Volume: 9
Electronic Location ID: e10710
Received 2020 Sep 24; Accepted 2020 Dec 15
Copyright: ©2021 Spaan et al.
Copyright year: 2021
Copyright holder: Spaan et al.
License: This is an open access article distributed under the terms of the Creative Commons Attribution License, which permits unrestricted use, distribution, reproduction and adaptation in any medium and for any purpose provided that it is properly attributed. For attribution, the original author(s), title, publication source (PeerJ) and either DOI or URL of the article must be cited.
License URL: https://creativecommons.org/licenses/by/4.0/

Keywords: Bighorn sheep, Juvenile survival, Mycoplasma ovipneumoniae, Population genetic diversity, Forage suitability, Targeted removals

Funding: United States Department of Interior and Oregon Department of Fish and Wildlife F15AF01356 F19AF00469 Oregon Foundation for North American Wild Sheep Nevada Bighorn Unlimited—Midas Nevada Bighorn Unlimited—Reno Nevada Muley’s This work was supported by the United States Department of Interior and Oregon Department of Fish and Wildlife (No. F15AF01356 and F19AF00469), Oregon Foundation for North American Wild Sheep, Nevada Bighorn Unlimited - Midas, Nevada Bighorn Unlimited - Reno, and Nevada Muley’s. The funders had no role in study design, data collection and analysis, decision to publish, or preparation of the manuscript.

==============================
Determining the demographic impacts of wildlife disease is complex because extrinsic and intrinsic drivers of survival, reproduction, body condition, and other factors that may interact with disease vary widely. Mycoplasma ovipneumoniae infection has been linked to persistent mortality in juvenile bighorn sheep (Ovis canadensis), although mortality appears to vary widely across subspecies, populations, and outbreaks. Hypotheses for that variation range from interactions with nutrition, population density, genetic variation in the pathogen, genetic variation in the host, and other factors. We investigated factors related to survival of juvenile bighorn sheep in reestablished populations in the northern Basin and Range ecosystem, managed as the formerly-recognized California subspecies (hereafter, “California lineage”). We investigated whether survival probability of 4-month juveniles would vary by (1) presence of M. ovipneumoniae-infected or exposed individuals in populations, (2) population genetic diversity, and (3) an index of forage suitability. We monitored 121 juveniles across a 3-year period in 13 populations in southeastern Oregon and northern Nevada. We observed each juvenile and GPS-collared mother semi-monthly and established 4-month capture histories for the juvenile to estimate survival. All collared adult females were PCR-tested at least once for M. ovipneumoniae infection. The presence of M. ovipneumoniae-infected juveniles was determined by observing juvenile behavior and PCR-testing dead juveniles. We used a known-fate model with different time effects to determine if the probability of survival to 4 months varied temporally or was influenced by disease or other factors. We detected dead juveniles infected with M. ovipneumoniae in only two populations. Derived juvenile survival probability at four months in populations where infected juveniles were not detected was more than 20 times higher. Detection of infected adults or adults with antibody levels suggesting prior exposure was less predictive of juvenile survival. Survival varied temporally but was not strongly influenced by population genetic diversity or nutrition, although genetic diversity within most study area populations was very low. We conclude that the presence of M. ovipneumoniae can cause extremely low juvenile survival probability in translocated bighorn populations of the California lineage, but found little influence that genetic diversity or nutrition affect juvenile survival. Yet, after the PCR+ adult female in one population died, subsequent observations found 11 of 14 ( 79%) collared adult females had surviving juveniles at 4-months, suggesting that targeted removals of infected adults should be evaluated as a management strategy.

Introduction

The study of population dynamics is essential for managing species (Williams, Nichols & Conroy, 2002). Recruitment is a crucial process for population dynamics, whereby populations gain individuals through births and immigration (Pradel, 1996). Recruitment varies strongly across species depending on whether they are k-selected, i.e., having fewer young with greater parental investment, or r-selected, i.e., having more young with reduced parental investment (MacArthur & Wilson, 1967). For k-selected species such as large terrestrial herbivores, annual adult survival tends to be relatively high with little variation. In contrast, juvenile survival tends to be more variable, and therefore population growth tends to be more sensitive to juvenile survival parameters (Gaillard et al., 2000). Consequently, it is essential to consider variables affecting the survival of juveniles when managing such species.

Bighorn sheep (Ovis canadensis) have a relatively low reproductive output (Festa-Bianchet et al., 2019). Females rarely have more than one offspring per year and may not achieve full reproductive potential until their 4th year (Rubin, Boyce & Bleich, 2000). Thus, juvenile survival can have a significant impact on population trajectories (Manlove et al., 2019). Juvenile survival is influenced by disease (Cassirer & Sinclair, 2007; Smith et al., 2014), maternal body condition (Festa-Bianchet, 1998), forage quality (Feder et al., 2008), weather (Douglas, 2001), genetic diversity (Hogg et al., 2006) and predation (Rominger, 2018). Disease, particularly pneumonia, can have a dramatic effect on juvenile survival (Manlove et al., 2016; Garwood et al., 2020). More broadly, respiratory disease likely caused the decline of bighorn sheep across western North America and continues to inhibit the recovery of the species (Cassirer et al., 2018). Therefore, evaluating the interaction of disease and other factors acting simultaneously on juvenile survival is critical for the conservation and management of bighorn sheep.

Mycoplasma ovipneumoniae, a bacterial pathogen, is considered the primary causative agent of respiratory pneumonia in bighorn sheep (Besser et al., 2008; Cassirer et al., 2018). Transmission of M. ovipneumoniae from domestic sheep (O. aries) and goats (Capra hircus) to bighorn sheep is typically followed by high mortality of individuals in all age classes in non-immune populations. Some survivors will clear the disease, while others remain chronic carriers that continue to shed the pathogen despite often appearing relatively healthy (Besser et al., 2013). Chronic carriers thus can transmit M. ovipneumoniae and sustain its presence within a population, especially by transmission to previously unexposed juveniles (Plowright et al., 2013). This infection pattern in bighorn sheep has been observed in multiple regions, including northeastern Oregon, Idaho, Washington, Nevada, and South Dakota (Cassirer et al., 2018; Garwood et al., 2020). Immunity to M. ovipneumoniae is thought to be strain-specific, with novel strains resulting in new waves of infection (Cassirer et al., 2017).

Population-level responses to M. ovipneumoniae outbreaks, however, may vary widely. Considerable variation has been observed in levels of all-age mortality at first contact, subsequent adult survival, and juvenile survival in following years among populations and across evolutionary lineages and habitats (Cassirer et al., 2018; Dekelaita et al., 2020). That variation has been hypothesized to stem from numerous causes, including strain virulence (Kamath et al., 2019), nutritional factors such as forage quality and population density (Dekelaita et al., 2020), stochastic factors such as the presence of chronic carriers (Cassirer et al., 2018; Garwood et al., 2020), genetic diversity of host populations (Cassirer et al., 2018), and phenological differences resulting in different patterns of aggregation, contact, and dispersal (Cassirer et al., 2018). Indeed, bighorn sheep inhabit ecosystems ranging from the arid deserts of northern Mexico and the southwestern United States of America to the frigid northern Rocky Mountains of Alberta and exhibit significant phenotypic variation and evidence of local adaptation (Wehausen & Ramey II, 2000; Wiedmann & Sargeant, 2014; Malaney et al., 2015). Currently, three subspecies of bighorn sheep are recognized, the desert bighorn sheep (O. c. nelsoni), the Rocky Mountain bighorn sheep (O. c. canadensis), and the Sierra Nevada bighorn sheep (O. c. sierrae) (Wehausen & Ramey II, 2000; Wehausen, Bleich & Ramey II, 2005). Previously, other subspecies were recognized (e.g., Peninsular bighorn sheep, O. c. cremnobates, and California bighorn sheep, O. c. californiana) (Cowan, 1940). Debate remains about whether those putative lineages reflect important independent evolutionary trajectories and important local adaptation (Buchalski et al., 2016; Bleich, Sargeant & Wiedmann, 2018; Barbosa et al., in press).

Previous studies of population dynamics in the presence of M. ovipneumoniae suggest that disease dynamics should be evaluated across lineages and ecosystems. For instance, Rocky Mountain bighorn sheep in the Hells Canyon system of Idaho, Oregon, and Washington occupy relatively continuous habitat with cold winters (Cassirer & Sinclair, 2007). In this system, M. ovipneumoniae tends to persist for long periods, resulting in constant disease in juveniles (Cassirer et al., 2018). The same pattern of infection was also observed in Rocky Mountain bighorn sheep in the Black Hills of western South Dakota and eastern Wyoming (Smith et al., 2014; Garwood et al., 2020). In the Mojave Desert of the southwestern United States of America, desert bighorn sheep occupy pockets of isolated mountain ranges surrounded by low-lying desert but linked by inter-mountain movements, resulting in natural metapopulations (Bleich et al., 1996). M. ovipneumoniae in this system appears to have been present periodically (Shirkley et al., in press), and impacts on adult survival and juvenile recruitment appear to be highly variable (Dekelaita et al., 2020).

Bighorn sheep in the northern Basin and Range ecosystem (including parts of southeastern Oregon, southwestern Idaho, and northern Nevada) occupy transitional habitats between the Mojave and Rocky Mountain systems and have a unique history. Native bighorn sheep from this region was considered the “California” subspecies (O. c. californiana; Cowan, 1940), spanned from the Sierra Nevada of California north to British Columbia. In Oregon, all native bighorn sheep were extirpated by 1945 (Oregon Department of Fish and Wildlife, 2003). Subsequently, morphometric and DNA analysis of the extinct native Oregon populations resulted in those populations being reassigned to the Great Basin desert form of the desert bighorn sheep (O. c. nelson; Wehausen & Ramey II, 2000). Bighorn sheep now existing in Oregon all stem from translocations and are managed as two lineages: Rocky Mountain bighorn sheep were introduced to northeastern Oregon, and “California” bighorn sheep were introduced to potential or former bighorn sheep habitat in other areas of the state using individuals from British Columbia. Both lineages are formally considered Rocky Mountain bighorn sheep subspecies at this time (O. c. canadensis; Wehausen & Ramey II, 2000). Most of the restored populations in Oregon were sourced from a single translocation of 22 bighorn sheep from British Columbia in 1953 (Olson, Whittaker & Rhodes, 2013). However, in 2000 and 2001, as part of an experimental effort to increase population genetic diversity and improve demographic performance, two populations in southeastern Oregon received augmentations of translocated bighorn sheep ultimately derived from different source populations in British Columbia (Olson, Whittaker & Rhodes, 2012). “California” bighorn sheep were also introduced to northern Nevada in 1972 from multiple source populations (NDOW, 2001; Olson, Whittaker & Rhodes, 2013), and dispersal from these populations into southeastern Oregon has been observed (ODFW, unpublished data). “California” bighorn populations in southwestern Idaho and northern Nevada share similar histories, although translocations to northern Nevada relied on a larger number of source populations than Oregon.

Bighorn sheep habitat in the northern Basin and Range ecosystem exhibits a metapopulation-like structure, where bighorn uses discrete patches of steep escape terrain separated by vast areas of grassland or sagebrush. Due to their demographic history and spatial distribution, bighorn sheep in southeastern Oregon and northern Nevada have low genetic diversity (Olson, Whittaker & Rhodes, 2013; Malaney et al., 2015) compared to levels measured in studies of Rocky Mountain bighorn sheep in the Rocky Mountains of Colorado (Driscoll et al., 2015) and desert bighorn sheep in the Mojave Desert of California (Epps, Crowhurst & Nickerson, 2018). Bighorn sheep in the northern Basin and Range ecosystem also experience different phenology of forage plants than observed in the desert or Rocky Mountain systems. Thus, disease dynamics may be expected to differ, as well.

Although cause-specific mortality in bighorn sheep juveniles has been widely studied in other systems (Smith et al., 2014; Cassirer et al., 2018; Cain et al., 2019; Garwood et al., 2020), it has not been evaluated in “California”-managed bighorn sheep. In this study, we evaluate the distribution and influence of M. ovipneumoniae on 13 “California”-managed bighorn sheep populations in southeastern Oregon and northern Nevada, and investigate juvenile mortality in relation to forage qualify and genetic diversity. Although presence of M. ovipneumoniae had previously been verified in at least one population, the effect of respiratory disease on the system was unknown. Genetic diversity of populations in this system likewise was unknown, although anticipated to be low, and forage quality was expected to vary widely given the broad range of elevations used by bighorn sheep and the strong influence of precipitation in this semi-arid system. We used telemetry and field-based observations to monitor the juveniles of GPS-collared adult females to estimate semi-monthly survival probability over a 3-year period. We hypothesized that survival of juveniles would be influenced by disease, nutrition, and genetic diversity. We predicted that the probability of 4-month juvenile survival would be lower in populations that (1) were exposed to M. ovipneumoniae, (2) had lower expected heterozygosity, and (3) that experienced lower forage quality, as indicated by pre- and post-parturition normalized differential vegetation index (NDVI). Additionally, after observing the mortality of an adult female suspected to be a chronic carrier in one of the study populations at the end of the 3-year period, we conducted a limited follow-up investigation of juvenile survival in that population in the following year.

Materials & Methods

Study area

The populations of bighorn sheep we studied were located in southeastern Oregon and northern Nevada, between 41.3 and 42.8°N, and 117.0 and 118.2°W (Fig. 1). The entire study area fell within the North Basin and Range (Level III classification of ecoregions in Omernik & Griffith, 2014). Five populations in our study fell within or largely within the Dissected High Lava Plateau (Blue Mountain (BSP), Bowden Hills (BHP), Rattlesnake (RSP), Three Forks (TFK), Upper Owyhee (UOP), Fig. 1), although BSP and RSP partly occurred within the High Lava Plains ecoregion (Omernik & Griffith, 2014). The Ten Mile (TMP) population fell within the High Lava Plains, while High Lava Plains and Semiarid Uplands (Omernik & Griffith, 2014) dominate bighorn sheep habitat within the Trout Creek (Trout Creek—east (TCE),—south (TCS) and—west (TCW), Fig. 1) and Santa Rosa metapopulation (Calicos (CAL), Eight Mile (EML), Martin Creek (MCK), and Sawtooth (SAW), Fig. 1).

Figure 1 Cumulative summer utilization distributions of adult female bighorn sheep populations considered in this study.

Overlapping polygons indicate shared habitat. Solid lined polygons, all to the west of U.S. Route 95, indicate populations unexposed to Mycoplasma ovipneumoniae, with dashed lines indicating exposed populations to the east of U.S. Route 95. Line fill indicates populations, EML and RSP, where a single adult female M. ovipneumoniae infection was detected. Green colored polygons indicate populations, BHP and RSP, where dead juveniles infected with M. ovipneumoniae were detected. Populations include—Bowden Hills (BHP), Blue Mountain (BSP), Calicos (CAL), Eight Mile (EML), Martin Creek (MCK), Rattlesnake (RSP), Sawtooth (SAW), Trout Creeks—east (TCE), Trout Creeks—south (TCS), Trout Creeks—west (TCW), Three Forks (TFK), Ten Mile (TMP) and Upper Owyhee (UOP).

The Dissected High Lava Plateau and High Lava Plains ecoregions are both characterized by elevated plateaus, but the Dissected High Lava Plateau contains sheer-walled canyons as well as intermittent lakes, while the High Lava Plains contains isolated volcanic cones and buttes as well as intermittent lakes and ephemeral streams (Omernik & Griffith, 2014). Mountains of low to mid elevation, typically with steep slopes, and some ephemeral and perennial streams characterize the Semiarid Uplands (Omernik & Griffith, 2014). The most common geology types across all three ecotypes are basalt and rhyolite, interspersed with other rock types. The soils derived from these rock types are fairly shallow and poor (Omernik & Griffith, 2014). Mean precipitation across the study area is typically 22.5–35.0 cm per year, although some areas of the Trout Creeks and Santa Rosa Mountains, receive significantly more precipitation (Omernik & Griffith, 2014).

All three ecotypes contain sagebrush steppe. Big sagebrush (Artemisia tridentata) and low sagebrush (A. arbuscular) are the most common woody herbaceous species, while common indigenous herbaceous species being primarily made up of palatable perennial bunchgrasses such as Idaho fescue (Festuca idahoensis), bluebunch wheatgrass (Psudoroegneria spicata), bottlebrush squirreltail (Elymus elymoides), Thurber needlegrass (Achnatherum thurberianum) and the less palatable Sandberg bluegrass (Poa secunda). Western juniper (Juniperus occidentalis) is the most common woody plant across 3 ecotypes, typically found in rocky areas, while the Semiarid Uplands are distinguished by willows (Salix spp.) in riparian areas, and quaking aspen (Populus tremuloides) and mountain mahogany (Cercocarpus spp.) in snow pockets (Omernik & Griffith, 2014). Ungulate species occurring in the study area include elk (Cervus canadensis), mule deer (Odocoileus hemionus), and pronghorn (Antilocapra americana) (Omernik & Griffith, 2014). Potential bighorn sheep predators in the study area include cougar (Puma concolor), coyotes (Canis latrans), bobcat (Lynx rufus), and golden eagles (Aquila chrysaetos; Omernik & Griffith, 2014).

The most common land-use practices are cattle ranching and cultivation of grains and hay (Omernik & Griffith, 2014). Heavy grazing of these lands and suppression of natural fires has led to the spread of large, uncontrollable fires and encroachment by invasive annual plants, such as the cheatgrass (Bromus tectorum) and medusahead (Taeniatherium caput-medusae; Omernik & Griffith, 2014). These grasses outcompete indigenous vegetation post-fire leading to the domination of these grasses. Wildlife and cattle here rely on springs, wetlands, and artificial water sources (Omernik & Griffith, 2014).

History of bighorn sheep populations within the study area

Reestablishment of bighorn sheep in the study area started in 1978 with the translocation of bighorn sheep into the Eight Mile area of the Santa Rosas (Fig. 1). The three Trout Creek populations, TCE, TCS, and TCW, and RSP were established with single translocations each from Hart Mountain (Table S1) in 1987 and 1992. Hart Mountain’s bighorn sheep population was established in 1954 with a translocation of bighorn sheep from Williams Lake, British Columbia. Although bighorn sheep in TMP and UOP were translocated from various other populations in Oregon, all those populations were ultimately derived from the population established at Hart Mountain in 1954. Bighorn sheep in CAL, although translocated from the Pine Forest Range in Nevada, are also derived from Hart Mountain. The remaining bighorn sheep populations in the study were established from either a different original source, e.g., MCK and SAW, which had stock from Kamloops, British Columbia, and Penticton, British Columbia, respectively, or more than one source, e.g., EML. Two populations were established by dispersal into unoccupied habitat: BSP was established with suspected dispersal events from the Trout Creek populations in the mid to late 1990s (pers. comm. S. Torland, ODFW), and BHP is thought to have been established by dispersing bighorn sheep from RSP. Other movements observed include the dispersal of collared adult males between the Santa Rosa Mountain Range and neighboring mountain ranges in southeastern Oregon, observed in 2009 and 2010 (NDOW, unpublished data).

Capture and sampling

All capture, handling, and disease testing were conducted by Oregon Department of Fisheries and Wildlife (ODFW) and Nevada Department of Wildlife (NDOW). Capture methodology followed the recommendations of Foster (2004) and the American Society of Mammalogists (Sikes & the Animal Care and Use Committee of the American Society of Mammalogists, 2016). ODFW and NDOW captured, collared, and sampled adult female bighorn sheep across 13 populations in southeastern Oregon and northern Nevada between January 2016 and February 2018 (Fig. 1). Captures were conducted using a net gun fired from a helicopter, with individual bighorn sheep blindfolded and hobbled once captured (Krausman, Hervert & Ordway, 1985). Bighorn were brought to a centralized area at the base of their range to be fitted with a telemetry collar and to collect biological samples, except where capture location was too far from basecamp to transport them quickly, in which case they were field processed, at the capture location.

Each adult female was fitted with a Vertex Survey Globalstar collar (Vectronic Aerospace, Berlin, Germany). These collars provide a GPS location every 13 h as well as VHF signal and were set to report a mortality if stationary for 12 h. Each collar had its own unique VHF frequency, with the occasional duplicate placed on individuals in different populations between which dispersal was deemed unlikely. The collars were also fitted with colored tags with unique numbers, allowing for identification of individuals observed in the field.

The age of each adult female was estimated from horn growth rings (Geist, 1966; Hoefs & Konig, 1984). Blood was obtained via jugular venipuncture to determine pregnancy status of adult females, obtain DNA, and to screen for disease. We determined pregnancy status of adult females using a serum pregnancy-specific protein B (PSPB) assay (Drew et al., 2001). Pregnancy testing of adult females was only conducted in the year in which they were captured; samples were sent to Sage Laboratories (Emmett, ID) to conduct testing.

Diagnostics

Presence of M. ovipneumoniae was detected with polymerase chain reaction (PCR) tests using nasal, bronchial, and tympanic bullae swabs from each captured female bighorn sheep (Manlove et al., 2019). Previous exposure to M. ovipneumoniae was determined using a competitive enzyme-linked immunosorbent assay (cELISA) to detect antibodies in serum (Ziegler et al., 2014). All tests for M. ovipneumoniae were performed at Washington Animal Disease Diagnostic Laboratory (WADDL).

Monitoring of juvenile bighorn sheep

From 2016 to 2018, we conducted semi-monthly observations of all collared adult females between April 1 and August 31. Juvenile identification was determined via observation of physical contact between adult females and juveniles, such as nursing or bedding down together. Juveniles are weaned at approximately 4-months of age (Festa-Bianchet, 1988); thus, our observation period was intended to cover birth through weaning. We located adult females for observation using an R-1000 telemetry receiver fitted with an RA-23K VHF directional antenna (Telonics, Inc., Mesa, AZ). We conducted observations with Kowa TSN-601 spotting scopes fitted with a 20–60x magnification mounted on tripods. Once an adult female was confirmed to not or no longer have a juvenile, i.e., two consecutive observations where the adult female was observed without a lamb, we stopped tracking that individual adult female (e.g., Cassirer & Sinclair, 2007).

We opportunistically located dead juveniles and collected samples, including either the entire corpse, the pluck (heart, liver, and lungs), the head, nasal and ear swabs, and/or tissue samples depending on the state of decomposition. Swabs were inserted into vials dry or containing tryptic soy broth, and other samples put in a cooler until returning to our field base. We then stored both the samples and swabs at −20 °C until laboratory submission. WADDL conducted gross- and histo-pathology on lung tissue samples and PCR tests on swabs. Additionally, M. ovipneumoniae strain-typing using multi-locus sequence typing (MLST) tests with four locus sequences, 16-23S intergenic spacer regions, the small ribosomal subunit, the genes encoding RNA polymerase B, and gyrase B, was conducted on lung tissue and swab samples from two of the juvenile mortalities recovered (Cassirer et al., 2017). We then compared these strain-types to strain-typed samples from the Santa Rosa metapopulation (n = 6) and from the Rattlesnakes (n = 2). WADDL conducted the M. ovipneumoniae strain-typing.

Genetic sampling

We used a combination of both blood samples (n = 125) and feces (n = 66) as sources for DNA samples. Whole blood (3 mL) provided by ODFW and NDOW from captured bighorn sheep was collected in EDTA tubes and spun at 4,000 × g for 10 min to separate the buffy coat. We extracted DNA from this material using a Qiagen DNeasy Blood and Tissue Kit (Qiagen Inc., Valencia, CA, USA). Fecal samples were collected opportunistically while conducting observations of bighorn sheep across the different populations, and were generally a week or less in age, as estimated from pellet color, odor, and surface condition. Fecal samples that were still moist after deposition were dried and then stored at room temperature. Fecal pellets were scraped to target dried epithelial cells on the surface of the pellet (Wehausen, Ramey & Epps, 2004), and we extracted DNA from the scraped material using a modified version of the Aquagenomic Stool and Soil protocol (Multitarget Pharmaceuticals LLC, Colorado Springs, CO; see details in Appendix S4).

Genotyping, markers, individual identification, and marker

We used a suite of 16 microsatellite markers in three panels (Table S2) that had previously been used to investigate population connectivity and genetic variability in bighorn sheep (Creech et al., 2020; Creech et al., 2020; Epps, Crowhurst & Nickerson, 2018). Genotyping followed protocols outlined in Epps, Crowhurst & Nickerson (2018). Briefly, all samples were run in at least two (for blood) or three (for feces) independent PCR reactions to generate a consensus genotype for each individual at each locus. For blood samples, any discrepancy between the two replicates resulted in the sample being rerun at that panel, although consistency across replicates was very high. Because allelic dropout can be higher in fecal samples, for those samples a homozygous genotype was considered verified if the single allele was seen in all three replicates. A heterozygous genotype was considered to be verified if each allele was seen in at least two of the three replicates; any other discrepancies resulted in reruns. Other studies on bighorn (e.g., Epps, Crowhurst & Nickerson, 2018) reported screening for recaptures using as few as 6 loci to achieve a desired probability of identity (PID; Waits, Luikart & Taberlet, 2001) of <0.001 and a probability of identity for full siblings of <0.05. However, our initial genotyping demonstrated that we needed to genotype at all 16 loci to achieve those thresholds. We identified recaptured individuals using cervus (Kalinowski, Taper & Marshall, 2007) by screening for individuals that matched at all 16 loci and removing them from the data set. We repeated this analysis using successively reduced numbers of matching loci and the presence of one to two mismatches (to account for missing data and genotyping error, respectively), until the matches that the program returned seemed unlikely due to geographic location and/or the mismatches were not explainable by simple allelic dropout. Finally, we used gimlet (Valière, 2002) to calculate two types of error rates in our genotypes: allelic dropout, and the presence of false alleles.

Linkage disequilibrium, Hardy-Weinberg tests, and genetic diversity

We used genepop Version 4.2 on the Web (Rousset, 2008) to conduct the probability test for Hardy-Weinberg equilibrium (HWE) for each population by locus and then for each locus by population, as well as across populations for each locus and across loci for each population (Fisher’s method, Fisher 1948). We then used genepop 4.0 Desktop to test for linkage disequilibrium across each pair of loci within each population, and each pair of loci across all populations, applying a sequential Bonferroni correction in both cases across all loci. We used the R package diveRsity (Keenan et al., 2013) to calculate population genetic diversity metrics including expected heterozygosity (HE), observed heterozygosity (HO), and allelic richness (AR). We accounted for imbalanced sample sizes amongst populations using rarefaction.

NDVI data

We used 14-day composite, 250 m resolution NDVI data from the Moderate Resolution Imaging Spectroradiometer (eMODIS). We utilized pre-processed data from 2016-2018 obtained from Earth Explorer (https://earthexplorer.usgs.gov/), which is managed by the United States Geological Survey’s Earth Resources Observation Center (Jenkerson, Maiersperger & Schmidt, 2010).

We used GPS data from all collared adult females in each population to generate a single utilization distribution per population for each year from 2016–2018, using the R package adehabitatHR (Calenge, 2006). We estimated 95% utilization distributions using the kernel method with the default smoothing parameter (href). We did not use least square cross validation (hlscv) due to repeated use of locations, which may cause convergence issues (Kie et al., 2010).

We then extracted NDVI data from each population polygon for each 14-day composite image and generated a 90th percentile statistic using program R (R Core Team, 2019). Bighorn sheep are selective feeders; as such, we assume that the 90th percentile NDVI statistic represents a high-quality choice of forage, while accounting for the fact that maximum forage is not always attainable (Creech et al., 2016). Additionally, selecting the 90th percentile excludes the selection of false maximum values caused by measurement error. Finally, we averaged NDVI values across each 14-day composite for the 3-months before and after the first juvenile observed in each population and used this variable as a measure for forage quality for each population pre- and post-parturition.

Drivers of juvenile survival

We analyzed our known-fate data in Program Mark (White & Burnham, 1999), to estimate juvenile survival (S) using a Kaplan–Meier estimator (Kaplan & Meier, 1958) with staggered entry (Pollock et al., 1989). First, we considered three population-level measures of exposure to M. ovipneumoniae in our models of juvenile survival (Table 1). Primarily, we considered the presence of M. ovipneumoniae-infected juveniles in each population, as the limited numbers of adults captured and tested in each population and the lack of annual testing precluded clear estimates of infection rates among adults. However, we also considered whether presence of infected adults (PCR) or exposed adults (cELISA) in each population influenced juvenile survival. Second, we considered two population-level measures of genetic diversity, expected heterozygosity (HE) and allelic richness (AR) (Table 1). We used univariate models of the three M. ovipneumoniae measures and the two genetic diversity measures as initial screening methods, using Akaike’s information criterion, corrected for small sample sizes (AICc) to determine which M. ovipneumoniae and genetic diversity metric most strongly linked to juvenile survival before proceeding with subsequent analyses.

Table 1 Description of variables considered in known-fate models predicting survival of juvenile bighorn sheep (Ovis canadensis) in populations across southeastern Oregon and northern Nevada.

All statistical measures were considered at the population level.

Measure	Category	Measure type	Statistical measure(range)	
. (null)		Intercept only model		
Time	Temporal	Time-varying		
T	Temporal	Linear time trend		
M. ovipneumoniae status	Bacteria	M. ovipneumoniae status (+/-), as determined by presence of infected juveniles.	Binary	
M. ovipneumoniae status	Bacteria	M. ovipneumoniae status (+/-) as determined by presence of infected (PCR+) adults	Binary	
M. ovipneumoniae status	Bacteria	M. ovipneumoniae status (+/-) as determined by presence of infected (cELISA+) adults	Binary	
AR (Allelic richness)	Genetic	Measure of genetic diversity	Continuous (values btw. 1.82–2.88)	
HE (Expected heterozygosity)	Genetic	Measure of genetic diversity	Continuous (values btw. 0–1)	
Pre-NDVI (Pre-parturition NDVI)	Nutrition	90th percentile 3-month pre-parturition mean NDVI	Continuous (values btw. −0.2–1)	
Post-NDVI (Post-parturition NDVI)	Nutrition	90th percentile 3-month post parturition mean NDVI	Continuous (values btw. −0.2–1)	

Subsequently, we conducted two analyses of juvenile survival. The first included all study populations (n = 13) and the selected measures of M. ovipneumoniae presence in each population, population-level metrics of forage quality (3-month pre- and post-parturition NDVI values) and genetic diversity (expected heterozygosity, HE) (Table 1), and both additive and multiplicative effects of time. For each model that included the multiplicative effect of time and M. ovipneumoniae, we fixed survival interval 1 for the M. ovipneumoniae group, as no mortalities occurred during this period. The second analysis included all of the same variables except M. ovipneumoniae presence and was restricted to populations where M. ovipneumoniae-infected juveniles were not detected (n = 11, see Results). That second analysis was undertaken to determine whether effects of M. ovipneumoniae obscured the effect of the other covariates of interest. For both analyses, we used AICc and AICc weights (wi) to select the best-supported model. We included a null model in both model selection sets to evaluate model performance (Burnham & Anderson, 2002). We selected the model with the lowest AICc and highest wi as our best-supported model. We used evidence ratios between the top model and competitive models (those within 2 AICc units, to evaluate each model relative to the top model (Burnham & Anderson, 2002).

Post-study observations

After our final planned field season in 2018, the single PCR+ adult female in the RSP (of 21 tested) died of suspected bluetongue (Orbivirus spp.). Because prevalence of PCR+ adult females in this population was low (4.76% of tested—see Results), and no adult males tested PCR+ throughout the study, we considered it possible that no additional PCR+ individuals remained at RSP, potentially removing the source of infection for new juveniles. Therefore, we decided to conduct a single observation of juveniles in the RSP and BHP, populations in early August of 2019; we included BHP given its proximity to RSP. Juveniles in both these populations were typically 4-months old at that time; thus, that observation aligned closely with the 4-month juvenile survival we estimated during the study.

Results

Diagnostics

Between 2016 and 2018, 78 adult females were tested via cELISA to determine M. ovipneumoniae exposure, and 95 adult females were tested via PCR to determine active M. ovipneumoniae infections. The proportion of adult females PCR tested in each population varied from 0.10 to 0.43 (x¯=0.27, Table S4). For the 10 adult females in the Santa Rosa metapopulation that were recaptured during our study and retested for infection via PCR, PCR status remained the same for the single positive individual in EML, and the rest were negative. None of the adult females in populations west of U.S. Route 95 (BSP, TCE, TCS, or TCW) showed evidence of M. ovipneumoniae exposure (Fig. 1, Table S3). However, all of the populations east of U.S. Route 95 (the four populations in the Santa Rosa Mountain’s metapopulation, as well as BHP, RSP, TMP, and UOP (Fig. 1)), except TFK, included adult females with evidence of exposure to M. ovipneumoniae (Table S3). We were unable to get a sample from the single individual adult female captured in TFK. The proportion of M. ovipneumoniae exposed adult females in those eight populations varied between 0.60 in CAL and EML to 1.00 in TMP (x¯=0.75; Table S4). Only two adult females tested PCR+ to M. ovipneumoniae infection across all 13 populations (Table S3): one in EML on two occasions (2017 and 2018), and one in RSP in 2016. Additionally, one adult female in BHP yielded an indeterminate PCR test result, meaning M. ovipneumoniae detection could not be determined (Besser et al., 2019).

Linkage disequilibrium, Hardy-Weinberg tests, and genetic diversity

After removing the fecal samples from recaptured individuals (n = 28), the genetic data set contained 191 individuals at 16 microsatellite loci, representing 10 to 26 (x¯=15.9) individuals per population (Table S5). The mean rate of false allele occurrence per locus was 0.001 and the mean allelic dropout rate across all loci was 0.008 (range = 0.000–0.017). No locus was determined to be out of Hardy-Weinberg equilibrium by either test employed (Tables S6 & S7). Evaluating each locus pair by population showed no evidence of linkage disequilibrium (pcritical = 0.000035 for α = 0.05); evaluating linkage for each pair of loci across populations using Fisher’s test indicated that BL4 and HH62 were in disequilibrium (p = 0.00039; pcritical = 0.00042 for α = 0.05). However, those loci have not appeared to be in disequilibrium in other bighorn sheep studies (e.g., Epps, Crowhurst & Nickerson, 2018), suggesting that this relationship may have been an artifact. Thus, and because linkage disequilibrium is more likely to bias estimates of genetic structure rather than estimates of genetic diversity as employed in this study, we retained all 16 loci in our analyses. HE across the 13 study populations varied from 0.26 in BSP to 0.48 in EML (x¯=0.37; Table S5) and AR varied from 1.82 in BSP to 2.88 in SAW (x¯=2.37; Table S5).

NDVI data

Pre- and post-parturition NDVI varied spatially and temporally (Table S8). BSP had consistently low pre-parturition NDVI, whilst EML and TMP had consistently high pre-parturition NDVI (Table S8). Post-parturition NDVI was lowest in BHP in 2018, was consistently low in RSP but was consistently high in TCE and EML (Table S8). Pre- and post-parturition NDVI values were not correlated (Table S9).

Pregnancy rates and observation of juvenile bighorn sheep

Seventy-six of the 82 (93%), pregnancy tests conducted across all 13 populations between 2016 and 2018 were positive (Table S10). Six of the 82 tests conducted were on recaptured adult females from EML (n = 3) and SAW (n = 3), all of which were positive. All populations had 100% pregnancy rates, except for TCE (67%; n = 8∕12), TCS (50%; n = 1∕2), and TFK (0%; n = 0∕1) in 2016. The single collared adult female in TFK did not yield a positive pregnancy test result in January, but was later observed in early June with a juvenile, suggesting that the test was administered too early to detect a positive pregnancy result. Sixty-five of those 82 (79%) pregnant adult females survived to parturition; of those 59 (91%) were observed with juveniles. Population pregnancy rates were not correlated with genetic diversity (Pearson pairwise correlation; HE, r = 0.24, p = 0.272; AR, r = 0.14, p = 0.529).

We observed 121 juveniles with radio-collared adult females between 2016 and 2018; 78% of radio-collared adult females were observed with juveniles. Populations with the lowest proportion of radio-collared adult females with juveniles were BSP (2016—0.00; 2017—0.67; 2018—0.33), and TCE (2016—0.64; 2017—0.64; 2018—0.17), although sample sizes were small in BSP (n = 3). A set of twins was observed with a radio-collared adult female in both the BHP and MCK populations in 2018.

The observation rate of collared adult females with juveniles varied from 50 to 100%, with a mean observation rate of 94% across all semi-monthly sampling periods and study populations (Table S11). Throughout the study, we collected samples suitable for M. ovipneumoniae testing from 17 juvenile mortalities (Table S12). All juvenile mortalities (n = 15) recovered from the BHP (n = 1) and RSP (n = 14) populations tested positive for M. ovipneumoniae. None of the other juvenile mortalities from BSP (n = 1) or TCS (n = 1) tested positive for M. ovipneumoniae. The M. ovipneumoniae strain-type of the samples collected from the BHP and RSP juveniles matched the strain-type (NV_BHS_SantaRosas_2651_2014_4; Kamath et al., 2019) of all the other previously strain-typed bighorn sheep at the available sequences from the Santa Rosa meta-population and Rattlesnake population (Table S13).

Drivers of juvenile survival

Preliminary analyses with univariate models of the relationship between juvenile survival and different population-level measures of M. ovipneumoniae revealed that the presence of M. ovipneumoniae-infected juveniles in the population was a much stronger predictor of juvenile survival than the presence of infected (PCR+) or exposed (ELISA+) adults (Table S14). Preliminary analyses with univariate models of juvenile survival as a function of genetic diversity demonstrated that HE was a stronger predictor of juvenile survival than AR (Table S14). Therefore, in our subsequent multivariate models of juvenile survival, we used the presence of M. ovipneumoniae-infected juveniles in each population as our measure of M. ovipneumoniae presence, and HE as our measure of genetic diversity. In those multivariate analyses, we identified two competing models predicting survival probability of juveniles across all populations (Table 2). Both models contained only two predictors: (1) whether M. ovipneumoniae-infected juveniles were detected, and (2) time. The model containing time as a multiplicative effect with M. ovipneumoniae had 2.36 times more support than the competing model that treated time as an additive effect (Table 2), meaning that the temporal pattern of juvenile mortality differed in populations where M. ovipneumoniae-infected juveniles were present. Neither forage quality (pre- and post-parturition NDVI), nor genetic diversity (HE) predicted survival probability of juveniles (Table 2).

Table 2 Model selection results for known fate models predicting cumulative 4-month survival of juvenile bighorn sheep (Ovis canadensis) in southeastern Oregon and northern Nevada for the period 2016–2018.

For analysis 1, we included a binary group variable for populations where Mycoplasma ovipneumoniae was (n = 2) and was not detected (n = 11). We modeled time as a constant (.), time-varying (time), linear (T), and random effect. Covariates modeled include expected heterozygosity (HE), and pre- and post-parturition NDVI. For analysis 2, we only modeled populations (n = 11) with no observed M. ovipneumoniae mortalities. We modeled time as a constant (.), time-varying (time), and linear (T) effect. We included the same covariates in data set 2 that we used in data set 1.

Analysis	Model	K	AICc	ΔAICc	wi	ML	
1	M. ovipneumoniae× time	15*	366.51	0.00	0.30	1.00	
	M. ovipneumoniae + time	9	368.23	1.71	0.13	0.42	
	M. ovipneumoniae× time + HE	16*	368.58	2.06	0.11	0.36	
	M. ovipneumoniae× time + post-NDVI	16*	368.63	2.12	0.10	0.35	
	M. ovipneumoniae× time + pre-NDVI	16*	368.64	2.12	0.10	0.35	
	M. ovipneumoniae + time + HE	10	370.02	3.51	0.05	0.17	
	M. ovipneumoniae + time + pre-NDVI	10	370.23	3.72	0.05	0.16	
	M. ovipneumoniae + time + post-NDVI	10	370.31	3.79	0.04	0.15	
	M. ovipneumoniae× time + HE + post-NDVI	17*	370.66	4.15	0.04	0.13	
	M. ovipneumoniae× time + HE + pre-NDVI	17*	370.69	4.18	0.04	0.12	
	M. ovipneumoniae× time + pre-NDVI + post-NDVI	17*	370.77	4.25	0.04	0.12	
	M. ovipneumoniae× time + HE + pre-NDVI + post-NDVI	18*	372.80	6.29	0.01	0.04	
	Time + post-NDVI	9	383.02	16.51	0.00	0.00	
	Time + HE	9	387.94	21.42	0.00	0.00	
	Time	8	392.21	25.70	0.00	0.00	
	Time + pre-NDVI	9	394.22	27.71	0.00	0.00	
	M. ovipneumoniae	2	400.46	33.94	0.00	0.00	
	post-NDVI	2	412.59	46.08	0.00	0.00	
	HE	2	417.59	51.07	0.00	0.00	
	T	2	418.76	52.24	0.00	0.00	
	. (null)	1	422.68	56.17	0.00	0.00	
	pre-NDVI	2	424.41	57.90	0.00	0.00	
2	Time	8	210.79	0.00	0.23	1.00	
	. (null)	1	211.84	1.05	0.14	0.59	
	Time + pre-NDVI	9	212.52	1.73	0.10	0.42	
	Time + HE	9	212.70	1.91	0.09	0.38	
	Time + post-NDVI	9	212.78	1.99	0.09	0.37	
	T	2	213.27	2.48	0.07	0.29	
	post-NDVI	2	213.60	2.82	0.06	0.24	
	pre-NDVI	2	213.67	2.88	0.05	0.24	
	HE	2	213.69	2.90	0.05	0.23	
	Time + HE + pre-NDVI	10	214.19	3.41	0.04	0.18	
	Time + pre-NDVI + post-NDVI	10	214.41	3.62	0.04	0.16	
	Time + HE + post-NDVI	10	214.78	4.00	0.03	0.14	
	Time + HE + pre-NDVI + post-NDVI	11	216.25	5.46	0.02	0.07	
Notes.

* Models where survival interval 1 for the M. ovipneumoniae group were fixed as no mortalities occurred during that interval

The odds of a juveniles surviving to 4-months of age in our study populations where M. ovipneumoniae-infected juveniles were detected were 8.00 (95% CI [−4.00–255.78]) times less likely than juveniles in populations where we did not detect M. ovipneumoniae-infected juveniles (Table 3). The derived probability of survival for the entire 4-month annual study period for juveniles in populations where M. ovipneumoniae-infected juveniles were detected was 0.02 (95% CI [0.00–0.13]) compared to 0.44 (95% CI [0.29–0.59]) for juveniles in other populations (Fig. 2B).

Table 3 The outputs from the top models in analysis 1 (M. ovipneumoniae× time) and analysis 2 (null) predicting survival of juvenile bighorn sheep (Ovis canadensis) in southeastern Oregon and northern Nevada.

Analysis 1 includes all study populations, while analysis 2 only includes populations (n = 11) where Mycoplasma ovipneumoniae was not detected.

Analysis	Covariate	Effect on survival	Odds-ratio	Estimate	SE	95% CI	
						Lower	Upper	
1	Intercept			2.08	1.06	0.00	4.16	
	M. ovipneumoniae	↓	8.00	−2.08	1.77	−5.54	1.39	
	Time 1	↑	9.50	2.25	1.46	−0.61	5.12	
	Time 2	↑	1.00	0.00	1.13	−2.22	2.22	
	Time 3	↓	1.89	−0.64	1.12	−2.84	1.56	
	Time 4	↓	1.05	−0.05	1.16	−2.33	2.23	
	Time 5	↑	1.83	0.61	1.22	−1.78	2.99	
	Time 6	↑	2.44	0.89	1.28	−1.63	3.41	
	Time 7	↓	1.09	−0.09	1.23	−2.49	2.32	
	M. ovipneumoniae× time 1*	↑	1.00	0.00	0.00	0.00	0.00	
	M. ovipneumoniae× time 2	↑	4.37	1.48	1.85	−2.16	5.11	
	M. ovipneumoniae× time 3	↑	4.36	1.47	1.85	−2.15	5.09	
	M. ovipneumoniae× time 4	↓	1.33	−0.29	1.88	−3.96	3.39	
	M. ovipneumoniae× time 5	↓	2.75	−1.01	1.97	−4.88	2.86	
	M. ovipneumoniae× time 6	↓	1.63	−0.49	2.12	−4.64	3.66	
	M. ovipneumoniae× time 7	↑	2.18	0.78	2.24	−3.60	5.16	
2	Intercept			2.08	1.06	0.00	4.16	
	Time 1	↑	9.50	2.25	1.46	−0.61	5.12	
	Time 2	↓	1.00	0.00	1.13	−2.22	2.22	
	Time 3	↓	1.89	−0.64	1.12	−2.84	1.56	
	Time 4	↓	1.05	−0.05	1.16	−2.33	2.23	
	Time 5	↑	1.83	0.61	1.22	−1.78	2.99	
	Time 6	↑	2.44	0.89	1.28	−1.63	3.41	
	Time 7	↓	1.09	−0.09	1.23	−2.49	2.32	
Notes.

* M. ovipneumoniae × time 1 is fixed as no mortalities were observed during this interval.

Figure 2 (A) Semi-monthly survival probabilities and (B) 4-month cumulative survival probabilities of juvenile bighorn sheep (Ovis canadensis) in populations across southeastern Oregon and northern Nevada.

(A) Semi-monthly survival probabilities and (B) 4-month cumulative survival probabilities of juvenile bighorn sheep (Ovis canadensis) estimated using a Kaplan–Meier estimator with staggered entry and known-fate data collected in 13 populations in southeastern Oregon and northern Nevada. Survival probabilities and 95% confidence intervals are for populations where Mycoplasma ovipneumoniae was and was not detected.

We observed no significant difference in semi-monthly juvenile survival probability between populations where M. ovipneumoniae-infected juveniles were not detected in observations at 0.5, 1, and 1.5-months post-parturition (Fig. 2A). However, semi-monthly survival probability in populations where M. ovipneumoniae-infected juveniles were present were significantly lower 2-months (S exposed = 0.42, 95% CI [0.24–0.62]; S unexposed = 0.88, 95% CI [0.75–0.95]) and 2.5-months (S exposed = 0.40, 95% CI [0.16–0.70]; S unexposed = 0.94, 95% CI [0.82–0.98]) post-parturition (Fig. 2A). At 3, 3.5, and 4-months post-parturition, estimates in semi-monthly survival probability again did not differ between populations where M. ovipneumoniae-infected juveniles were or were not detected (Fig. 2A). The increase in uncertainty in our estimates is tied to decreasing sample size as individuals died.

Although no direct effect of either population genetic diversity or post-parturition NDVI was supported based on our model selection approach, both variables were highly correlated with the presence of M. ovipneumoniae (HE = 0.58; post-NDVI = −0.66, Table S9). In fact, the two populations where M. ovipneumoniae-infected juveniles were present, BHP and RSP, had relatively high genetic diversity (HE: BHP = 0.43; RSP = 0.45) compared to other study populations (Table S5). Similarly, RSP had low post-parturition NDVI values relative to the other populations in 2017 and 2018, while the same was the case for BHP in 2018 (Table S8).

In the second analysis, which excluded the two populations where M. ovipneumoniae-infected juveniles were present, competing models included both indices of forage quality and population genetic diversity as predictors of juvenile survival (Table 2), as well as a time-varying effect. However, the 95% confidence intervals for the parameter estimates of the nutritional and population genetic diversity model both overlapped zero, and the null model with no predictors was 2nd ranked. The top competing model, a time-varying model, had 1.69 times more support than the next best model, the null model (Table 2).

Post-study observations

In early August of 2019, following the death of the single adult female in RSP that tested PCR+ for M. ovipneumoniae, we located the two remaining collared individuals in the BHP (linked to RSP by adult male and female movements, data not shown). Neither the collared adult females nor any of the eight other adult females we observed in this population had juveniles (0% juvenile survival). We then located 14 of the RSP collared adult females, of which 11 had juveniles (∼79% juvenile survival). We located approximately 35 adult females total in the RSP population and observed approximately 27 juveniles, which equates to an approximate 4-month survival rate of ∼77%.

Discussion

We found that M. ovipneumoniae had a strong negative effect on juvenile survival in populations where infected juveniles were detected. Juveniles in those populations had a 4-month survival probability more than 20 times lower than juveniles in populations where we did not detect infected juveniles. Indeed, in one affected population (RSP), only two juveniles out of 40 over three years survived. Presence of adult bighorn sheep with evidence of past exposure to M. ovipneumoniae, and in one population the presence of an adult infected at time of capture, were not strong predictors of juvenile survival. Moreover, juvenile survival was not detectably influenced by forage quality or population genetic diversity, even when excluding populations where M. ovipneumoniae-related juvenile mortality was observed. Strikingly, the death of a single infected adult female in a heavily impacted population appeared to drive an immediate recovery of juvenile survival in that population in the subsequent year, underscoring the potentially pivotal role of chronically infected adult females in outbreaks of this disease.

In our study, four-month juvenile survival probability in populations where infected juveniles were present (0.02; 95% CI [0.00–0.13]) were comparable to some of the lowest observed juvenile survival probabilities reported in other studies where pneumonia epizootics were present. In Rocky Mountain bighorn sheep in the Hells Canyon system of Idaho, Oregon, and Washington, survival to weaning at 4–5 months of age had a median value of 0.10, but ranged from <0.01–0.69, with peak mortality occurring between 1.5 and 2.5 months of age (Cassirer et al., 2013). In the Black Hills of South Dakota, Smith et al. (2014) reported 52-week survival to be only 0.02 (95% CI [0.01–0.07]), with the majority of juveniles (∼75%) dying within the first 2 months after parturition. We observed similar temporal patterns to those studies, with most mortality related to disease in our study occurring between 1.5 and 3-months after parturition. In the Mojave Desert of California, in a 3-year study of desert bighorn sheep following an outbreak of M. ovipneumoniae, the proportion of juveniles surviving varied from 0.00 to 0.92 in populations where at least five juveniles were monitored, and survival in most populations improved in later years (Dekelaita et al., 2020). Unlike Cassirer et al. (2013) and Smith et al. (2014), we did not address the effect of predators on juvenile survival. However, it should be noted that even if cause-specific mortality were to be assigned to a predator, the ultimate cause of mortality might still be M. ovipneumoniae. Although M. ovipneumoniae is considered the primary agent of pneumonia, other diseases, and the potential for co-infection effects may be important, but we did not assess them. Periods of low juvenile survival due to pneumonia can result in demographic consequences such as declining population growth rates (Manlove et al., 2016) and even skewed population structures (Festa-Bianchet, Gaillard & Côté, 2003).

Our study adds to the evidence that dynamics of respiratory pneumonia in bighorn sheep may be strongly influenced by the presence of only a small number of infected adults. We detected only one infected female adult bighorn sheep across two populations where M. ovipneumoniae-infected juvenile bighorn sheep were observed. However, the presence of infected adults at capture was not the strongest predictor of juvenile survival. For instance, in EML, an infected female adult was present in 2017; she was still infected when retested in 2018. Despite that, one out of four juveniles born in 2017 and five of six juveniles born in 2018 survived. It is unclear why the presence of this apparently chronically-infected female did not lead to juvenile mortality in this case, although Plowright et al. (2017) noted that juvenile survival was not directly related to the mother’s infection status in the study. The lack of juvenile die-off could be explained by other factors, e.g., the stochastic nature of exposure and the possibility of co-infection. We attempted to capture other factors that may play a role, e.g., measures of forage quality and population genetic diversity, but we did not detect any direct effect of those factors. Additionally, we were unable to assess the interaction of these variables with M. ovipneumoniae infection because our measures of those factors were at the population level. Conversely, juvenile survival in RSP was extremely poor from 2016–2018, where a single infected adult female was detected in 2016. Although we did not retest her and therefore do not know whether she was chronically infected, after her death in late 2018, juvenile survival rebounded sharply in 2019. In a satellite population discovered in the course of this study, BHP, one female out of three captured in 2018 yielded an indeterminate PCR result. We surmise that additional captures likely would have detected infected adults given that infected juveniles were detected in 2018 and no surviving juveniles were detected in either 2018 or 2019. Experimental removals of chronically-infected adult females in other bighorn sheep studies (e.g., Garwood et al., 2020) demonstrate that chronically-infected adult females can strongly influence juvenile survival when prevalence is low.

We did not detect influences of population genetic diversity on juvenile survival, although we observed low to very low population genetic diversity throughout our study area, with HE ranging from 0.26–0.48 (Table S5). In contrast, HE values derived for 11 native desert bighorn sheep populations in the Mojave Desert of California, using 15 of the 16 microsatellite loci we employed, ranged from 0.50–0.69 (Epps, Crowhurst & Nickerson, 2018). However, our study design did not account for possible interactions between these variables and presence of M. ovipneumoniae-infected juveniles, because M. ovipneumoniae-related juvenile mortalities were confirmed in only two of 13 populations. Our estimates of genetic diversity were also only derived from 16 microsatellite loci, most putatively neutral; estimates based on more markers across the genome or additional genes associated with immune function could have been more informative. Interestingly, the populations where M. ovipneumoniae-related juvenile mortality was confirmed had relatively high genetic diversity (Table S5). In fact, the higher genetic diversity observed at RSP likely resulted from gene flow from males dispersing from the Santa Rosa metapopulation (NDOW, unpublished data). Such movements also appeared to have spread this strain of M. ovipneumoniae from the Santa Rosa metapopulation to BHP and RSP; indeed, only one strain is known to occur throughout the populations where we detected infected individuals (Table S13). Genetic diversity in populations to the west of U.S. Route 95, in particular, was strikingly low (HE was 0.26–0.33) due to founder effects from translocation history (see above). Genetic diversity observed by Olson, Whittaker & Rhodes (2013) for the source population, Hart Mountain, was noticeably higher (HE = 0.42). We note that the pregnancy rate in one of those populations (TCE) likewise was strikingly low (67%, n = 8∕12) in 2016. Although the adult females that we had the opportunity to test in 2017 and 2018 (n = 3 total) were pregnant, the numbers of adult females observed with juveniles in 2017 and 2018 were equal to or lower than 2016. Yet, those populations were apparently free of M. ovipneumoniae during our study. Such low pregnancy rates suggest that inbreeding or low genetic diversity could influence performance of those populations (Olson, Whittaker & Rhodes, 2012), although we cannot rule out other factors. These contrasts highlight the challenges of maintaining genetic diversity while limiting spread of disease, particularly in reintroduced systems of populations where compounding founder effects have resulted in some of the lowest reported values for genetic diversity in wild bighorn sheep populations.

Forage quality as assessed by NDVI values pre- and post-parturition likewise did not appear to influence juvenile survival consistently across the study. Again, our ability to evaluate how variation in forage quality influenced survival when M. ovipneumoniae was present was limited. We note, however, that of the three populations where M. ovipneumoniae-infected adult females or juveniles were detected, juvenile survival was lowest where forage quality was lowest (BHP and RSP, Table S8, average juvenile survival = 0 and 0.07, respectively). In contrast, EML had the highest or second-highest pre- and post- parturition NDVI values (Table S8) and higher juvenile survival (x¯=0.54; range = 0.25–0.83). Taken together, these anecdotal patterns suggest that the link between forage quality and M. ovipneumoniae dynamics within infected populations bears further investigation.

Our study demonstrated that reintroduced populations of “California”-managed bighorn sheep in southeastern Oregon and northern Nevada have been strongly influenced by respiratory disease, but not all populations are currently affected. Where infected juveniles were present, rates of juvenile mortality that we observed would lead to a dramatic decline in populations if those patterns persisted, as observed by Manlove et al. (2016). However, the rapid improvement in juvenile survival in one badly-affected population following the death of the single adult female known to be infected with M. ovipneumoniae at the start of the study suggests that identification and removal of chronically-infected adult females could be a successful management strategy in this system, as tested for a Rocky Mountain bighorn sheep population by Garwood et al. (2020). The very low estimates of genetic diversity we observed in some populations suggest that actions to manage for improved genetic diversity may be warranted, although natural movements between several study populations simultaneously resulted in increased genetic diversity and spread of M. ovipneumoniae. Thus, any genetic management strategy employed would have to mitigate the potential impact on spread of disease.

Conclusions

We found that populations with M. ovipneumoniae-infected juveniles had significantly lower juvenile survival, while no direct effect of either genetic diversity or pre- and post-parturition forage quality on juvenile survival was detected. Presence of M. ovipneumoniae-infected or exposed adults was less predictive of juvenile survival. Our findings add to the body of evidence that M. ovipneumoniae can have deleterious effects on juvenile survival across the various bighorn sheep lineages and the different ecosystems they inhabit. After the mortality of the only PCR+ adult female that we detected in one of the populations most affected by respiratory disease during the first three years of our study, we found increased juvenile survival and detected no infected juveniles in the following year. This natural experiment suggests that in populations where prevalence of PCR+ adults is low, removal of infected adults or adults identified as chronic carriers of M. ovipneumoniae. However, our ability to detect those influences may have been limited by small number of populations where respiratory disease was observed and high correlations between genetic diversity and presence of disease resulting from translocation history and patterns of connectivity in this system. Future studies should evaluate the interaction of genetic diversity and disease in systems with larger numbers of populations affected by respiratory disease.

Supplemental Information

Supplemental Information 1 Raw data and outputs used in figures

Raw data utilized in Program Mark to generate results, and associated figures, and tables.

Click here for additional data file.

Supplemental Information 2 Supplemental material

Click here for additional data file.

We thank Phillip Milburn, Scott Torland and Autumn Larkins of Oregon Department of Fish and Wildlife (ODFW) for helping to initiate this project, and Ed Partee of Nevada Department of Wildlife (NDOW) for support when working with the neighboring Nevada populations. We thank ODFW and NDOW staff for the capture, collaring, and sampling of bighorn sheep, particularly veterinarians, Julia Burco (ODFW), Peregrine Wolff (NDOW), Brianna Beechler (Oregon State University), and their support staff. We also thank the essential contributions in data collection by field technicians Geoff Gerdes, Logan Gmuender, Lindsey Howard, and Colton Padilla. The findings and conclusions in this publication are those of the authors and should not be construed to represent any official U.S. Department of Agriculture or U.S. Government determination or policy. The use of trade or firm names in this publication is for reader information and does not imply endorsement by the U.S. Government of any product or service.

Additional Information and Declarations

Competing Interests

Author Contributions

Animal Ethics

Field Study Permissions

Data Availability

The authors declare there are no competing interests.

Robert S. Spaan conceived and designed the experiments, performed the experiments, analyzed the data, prepared figures and/or tables, authored or reviewed drafts of the paper, and approved the final draft.

Clinton W. Epps conceived and designed the experiments, prepared figures and/or tables, authored or reviewed drafts of the paper, and approved the final draft.

Rachel Crowhurst and Adam Duarte analyzed the data, prepared figures and/or tables, authored or reviewed drafts of the paper, and approved the final draft.

Donald Whittaker and Mike Cox conceived and designed the experiments, authored or reviewed drafts of the paper, and approved the final draft.

The following information was supplied relating to ethical approvals (i.e., approving body and any reference numbers):

All capture, handling, and disease testing were conducted by Oregon Department of Fisheries and Wildlife (ODFW) and Nevada Department of Wildlife (NDOW). Capture methodology followed the recommendations of Foster (2004) and the American Society of Mammalogists (Sikes & the Animal Care and Use Committee of the American Society of Mammalogists, 2016).

The following information was supplied relating to field study approvals (i.e., approving body and any reference numbers):

Collection of samples from juvenile bighorn sheep mortalities was approved by Oregon Department of Fish and Wildlife, and Nevada Department of Wildlife.

The following information was supplied regarding data availability:

Data utilized in known fate models, raw data, and outputs are also available in the Supplemental Files.

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
