# Peer review of "Impact of Mycoplasma ovipneumoniae on juvenile bighorn sheep (Ovis canadensis) survival in the northern Basin and Range ecosystem"

_PeerJ, doi:10.7717/peerj.10710_

## Round 0.1 · original submission · Minor Revisions

Both reviewers mentioned that the manuscript is suitable for publication, but advised you to perform additional proof editing to correct occasional typos and improved a few sentences as specified in their comments.

Reviewer 1 ·

Basic reporting

The article is generally well written, but could use additional careful copy editing to correct occasional typographical errors and unclear sentences as specified in the general comments to the author below. The introduction and background clearly demonstrate how the work fits into the broader field of knowledge. The article shows professional structure and appropriate figures and table and raw data. The submission is self-contained and is clearly an appropriate unit of publication.

Experimental design

Original primary research within the journal scope. Clearly defined research question. Rigorous investigation to a high technical standard, in conformity with appropriate ethical standards. Clearly defined methods.

Validity of the findings

Underlying data provided, robust sound and controlled. Conclusions well stated, linked to original research questions, and limited to supporting results.

Additional comments

General comments: The manuscript is well written and complete. The study addresses important issues, including the generalizability of the hypothesis of the central role of M. ovipneumoniae in epizootic pneumoniae in bighorn lambs to a new host management subtype and a new ecoregion, and the role of the frequently hypothesized co-factors host genetics and nutrition. The results support the generally very limited genetic diversity in ‘California’ bighorn sheep in both the M. ovipneumoniae exposed and non-exposed study populations, without clear impact on juvenile survival. Similarly, the NVDI analyses failed to identify a major role for nutrition among the study populations. This study presents clear hypotheses (tested with rigor despite what must have been very trying field observation conditions over a sustained period), with appropriate analyses and well-justified conclusions.
Specific comments:
L37 needs clarification which populations ‘those’ refers to.
L41 consider ‘presence’ or ‘exposure to’ instead of ‘persistence’? It’s not the persistence of the bacterium alone; it’s that transmission of the bacterium to the lambs caused fatal pneumonia?
L74 suggest ‘non-immune’. Per Cassirer 2017, previously exposed and infected bighorn sheep remain susceptible to disease triggered by transmission of a new strain.
L78 suggest ‘previously unexposed’
L80 Seems like Parr et al, 2018 studied a population that was not infected with M. ovipneumoniae, so I’m not sure it is relevant to this sentence. Seems to me that there are several reports in the proceedings of the NWSGC symposia that support this point. I’m not sure how many in the broader literature.
L81 The citation presents this hypothesis based on one outbreak, so perhaps ‘thought to be’ would be better than ‘tends’.
L110 same comment as for L80
L140 ‘than’ Oregon (?)
L195 Can the authors confirm the accuracy of the Bowden Hills polygons in Figure 1 (with one of the two seemingly overlapping the Rattlesnake population?
L266 No doubt the deeper sites reflect infection, but nasal carriage may be just (inactive) colonization. Suggest ‘Presence of M. ovipneumoniae was detected…’
L268 I believe the WADDL PCR test was modified subsequently. For the time period cited, the method was published in Manlove et al. Prev Vet Med 2019.
L270 ‘Washington Animal Disease Diagnostic Laboratory’
L319 ‘and’ ?
L356-7 I think I know what this sentence means, but please clarify if possible.
L366-71 It would be helpful if the authors reported their best estimates of the proportion of the adult ewes in each population that were sampled for PCR detection of M. ovipneumoniae.
L396-8 incomplete sentence?
L421 ‘and one in RSP’?
L422-4 Basically, an ‘Indeterminate’ result is a very weak signal which may or may not be due to the target organism. It does not mean that the target was detected, as stated here.
L437-9 Any point in a ‘sensitivity analysis’ to determine if not using these two loci changes anything?
L444 ‘…and post-parturition NVDI varied…?
L487-9 What does it mean that time as a multiplicative effect had more support than time as an additive effect?
L501-507 It would be helpful if the text indicated these were semi-monthly (not cumulative) figures.
L539-41 I would appreciate more discussion about the choice of ‘detection of M. ovipneumoniae infected juveniles’ as the marker for exposure. It could be seen as circular reasoning to conclude that juveniles dying with M. ovipneumoniae associated pneumonia is associated with decreased juvenile bighorn survival. It would be helpful if the authors could comment on this and some related questions: What percent of bighorns were sampled for nasal carriage, by population (to help assess the likelihood that carriers were present but not detected)? Is it known how much bighorn movement exists among the bighorn populations east of Hwy 95 (to help understand the presence of seropositive animals despite the lack of detected carriers in most of those populations)? How are the results in the EMP (with at least one M. ovipneumoniae infected ewe but no detected infected juveniles) reconciled with the point made regarding the potentially pivotal role of chronic carriers in L545-6. Any ideas about how the situation in EMP and also in the other east of Hwy 95 populations with evidence of exposure to M. ovipneumoniae could be replicated (to improve lamb survival in the face of one or more ch
L598-9 Also unpublished data? Or was this reported in Kamath et al. 2019? Was this same strain carried by the detected carrier in EMP?
Figure 1 Consider adding color or hatching coding to indicate populations with cELISA evidence of exposure to M. ovipneumoniae, and to indicate populations with PCR evidence of M. ovipneumoniae carrier presence separately from populations with evidence of M. ovipneumoniae infected juveniles.
In several of the legends associated with the figures and table seem to have formatting issues. Maybe artefacts of pdf generation?
Table S4 There seems to be a typo in the tallies for the RSP PCR population, totalling 0.05.
Table S10 There are three asterisks in the table, but only one is defined. Are the column headers correct? For example, does ‘# adult females at start of year’ refer to population totals (as it seems to read) or to just collared individuals.

Reviewer 2 ·

Basic reporting

The paper describes a clear and original study carried out on different populations of bighorn sheep ranging in different areas of Oregon and Nevada, investigating the impact of Mycoplasma ovipneumoniae and other possible factors (genetic variation, nutrition, population density etc) interacting with survival or mortality of the new generation of weaning Bighorn sheep.

Statistical analysis of results has given a very strong proof of the importance of the exposition to this pathogen as responsible for lamb mortality in all groups screened. Moreover only the DNA evidence on two adults out of n° 13 populations is very difficult to link to clinical pneumonia or outbreaks.

Nevertheless the paper concerns a large collection of ecological and veterinary information for the American “Northern Basin and Range ecosystem” and could be considered an additional contribution to this field, adding scientific data on “at risk” species very important for future conservation strategies.

Some notes to be considered:

Introduction- overlong with some repetition eg L 58-70 and 83-92

Difficult to follow because of huge compound nouns eg L24-25 Five word noun; L35-36 Four words; L36-37 6 word noun. Sounds like "jargon".

L26 Few readers would understand: California” lineage populations

L30 Really dificult to understand this sentence- please break it down so it is easier to comprehend

L33 not sure how accurate diagnosis of M ovipneumoniae by behaviour can be

L35-36 We detected M. ovipneumoniae infected juvenile mortalities- do the authors mean they found some young dead sheep positive to M ovipneumoniae?

L36-37 Very difficult to understand: seems to mean that young sheep survived better when there were no infected sheep in the flock

L41-43 Just seems to suggest that the presence of M ovipneumoniae in sheep populations can lead to death which is not surprising but once again “translocated “California” bighorn populations” won’t mean much to many readers

L44-46 I’m not sure you can conclude much from the loss of one female

L52 some explanation of “k” and “r” is necessary

L64 delete comma after “...2006)”

L74 Please explain “an all-age die-off” readers may not know this jargon. Also L84

Generally discussion could be shortened as some overlap

Experimental design

It’s very difficult to plan such a wide investigation on living, protected animals which is difficult to handle respecting ethical standards, however the study was well performed and described taking into account not only the presence of the pathogen but also some significant genetic and environmental parameters and delivering important information on Mycoplasma diseases risks and genetic variability on this re-established species in those areas.

Validity of the findings

There are very efficient approaches for analysis of survival data such as weibull or cox model that we don’t see in this work however all data have been always supported by robust statistical approach.

Additional comments

No comments

---

## Round 0.2 · accepted · Accept

You have taken all the comments and suggestions of our reviewers very seriously and performed really nice work on improving the manuscript.